# Sequential Matching Network: A New Architecture for Multi-turn Response Selection in Retrieval-based Chatbots

## Abstract

We study response selection for multi-turn conversation in retrieval based chatbots. Existing work either concatenates utterances in context or matches a response with a highly abstract context vector finally, which may lose relationships among the utterances or important information in the context. We propose a sequential matching network (SMN) to address both problems. SMN first matches a response with each utterance in the context on multiple levels of granularity, and distills important matching information from each pair as a vector with convolution and pooling operations. The vectors are then accumulated in a chronological order through a recurrent neural network (RNN) which models relationships among the utterances. The final matching score is calculated with the hidden states of the RNN. Empirical study on two public data sets shows that SMN can significantly outperform state-of-the-art methods for response selection in multi-turn conversation.

## 1 Introduction

Conversational agents include task-oriented dialog systems and non-task-oriented chatbots. Dialog systems focus on helping people complete specific tasks in vertical domains (Young et al., 2010), while chatbots aim to naturally and meaningfully converse with humans on open domain topics (Ritter et al., 2011). Existing work on building chatbots includes generation based methods and retrieval based methods. Retrieval based chatbots enjoy the advantage of informative and fluent responses, because they select a proper response for the current conversation from a repository with re-

| Context |
|---|
| *Human*: How are you doing? |
| *ChatBot*: I am going to **hold a drum class** in Shanghai. Anyone wants to join? The location is near Lujiazui. |
| *Human*: Interesting! Do you have coaches who can help me practice **drum**? |
| *ChatBot*: Of course. |
| *Human*: Can I have a free first lesson? |
| **Response Candidates** |
| *Response 1*: Sure. Have you ever played drum before? ✓ |
| *Response 2*: What lessons do you want? ✗ |

Table 1: An example of multi-turn conversation

sponse selection algorithms. While most existing work on retrieval based chatbots studies response selection for single-turn conversation (Wang et al., 2013) which only considers the last input message, we consider the problem in a multi-turn scenario. In a chatbot, multi-turn response selection takes a message and utterances in its previous turns as input and selects a response that is natural and relevant to the whole context.

The key to response selection lies in input-response matching. Different from single-turn conversation, multi-turn conversation requires matching between a response and a conversation context in which one needs to consider not only the matching between the response and the input message but also the matching between the response and the utterances in previous turns. The challenges of the task include (1) how to identify important information (words, phrases, and sentences) in the context that is crucial to selecting a proper response and how to leverage the information in matching; and (2) how to model relationships among the utterances in the context. Table 1 illustrates the challenges with an example. First, "hold a drum class" and "drum" in the context are very important. Without them, one may find responses relevant to the message (i.e., the last turn of the context) but nonsense in the context (e.g., "what lessons do you want?"). Second,

the message highly depends on the second turn in the context, and the order of the utterances matters in response selection: exchanging the third turn and the last turn may lead to different responses. Existing work, however, either ignores relationships among utterances when concatenating them together (Lowe et al., 2015), or loses important information in context in the process of converting the whole context to a vector without enough supervision from responses (e.g., by a hierarchical RNN (Zhou et al., 2016)).

We propose a sequential matching network (SMN), a new context based matching model that can tackle both challenges in an end-to-end way. The reason that existing models lose important information in the context is that they first represent the whole context as a vector and then match the context vector with a response vector. Thus responses in these models cannot meet the context until the final step in matching. To avoid information loss, SMN matches a response with each utterance in the context at the beginning and encodes important information in each pair into a matching vector. The matching vectors are then accumulated in the temporal order of the utterances to model their relationships. The final matching degree is computed with the accumulation of the matching vectors. Specifically, for each utterance-response pair, the model constructs a word-word similarity matrix and a sequence-sequence similarity matrix by the embedding of words and the hidden states of a recurrent neural network with gated recurrent unites (GRU) (Chung et al., 2014) respectively. The two matrices capture important matching information in the pair on a word level and a segment level respectively, and the information is distilled and fused as a matching vector through an alternation of convolution and pooling operations on the matrices. By this means, important information from multiple levels of granularity in the context is recognized under sufficient supervision from the response and carried into matching with minimal loss. The matching vectors are then uploaded to another GRU to form a matching score for the context and the response. The GRU accumulates the pair matching in its hidden states in the chronological order of the utterances in the context. It models relationships and dependencies among the utterances in a matching fashion and has the utterance order supervise the accumulation of pair matching. The match-

ing degree of the context and the response is computed by a logit model with the hidden states of the GRU. SMN extends the powerful "2D" matching paradigm in text pair matching for single-turn conversation to context based matching for multi-turn conversation, and enjoys the advantage that both important information in utterance-response pairs and relationships among utterances are sufficiently preserved and leveraged in matching.

We test our model on the Ubuntu dialogue corpus (Lowe et al., 2015) which is a large scale public English data set for research in multi-turn conversation. The results show that our model can significantly outperform state-of-the-art methods, and improvement to the best baseline model on $R_{10}@1$ is over $6\%$. In addition to the Ubuntu corpus, we create a human labeled Chinese data set, namely Douban Conversation Corpus, and test our model on it. Different from the Ubuntu corpus in which data is collected from a specific domain and negative candidates are randomly sampled, conversations in this data come from open domain, and response candidates in this data set are collected from a retrieval engine and labeled by three human judges. On this data, our model improves the best baseline model over $3\%$ on $R_{10}@1$ and $4\%$ on $P@1$. As far as we know, Douban Conversation Corpus is the first human labeled data set for multi-turn response selection and could be a good complement to the Ubuntu corpus. We release Douban Conversation Corups and our source code at *an anonymous url for blind review. We have uploaded code and data with this paper.*

Our contributions in this paper are three-folds: (1) proposal of a new context based matching model for multi-turn response selection in retrieval based chatbots; (2) publication of a large human labeled data set to research communities. (3) empirical verification of the effectiveness of the model on public data sets;

## 2 Related Work

Recently, building a chatbot with data driven approaches (Ritter et al., 2011; Higashinaka et al., 2014) has drawn a lot of attention. Existing work along this line includes retrieval based methods (Hu et al., 2014; Ji et al., 2014; Wang et al., 2015; Yan et al., 2016; Wu et al., 2016b; Zhou et al., 2016; Wu et al., 2016a) and generation based methods (Shang et al., 2015; Serban et al., 2015; Vinyals and Le, 2015; Li et al., 2015, 2016; Xing

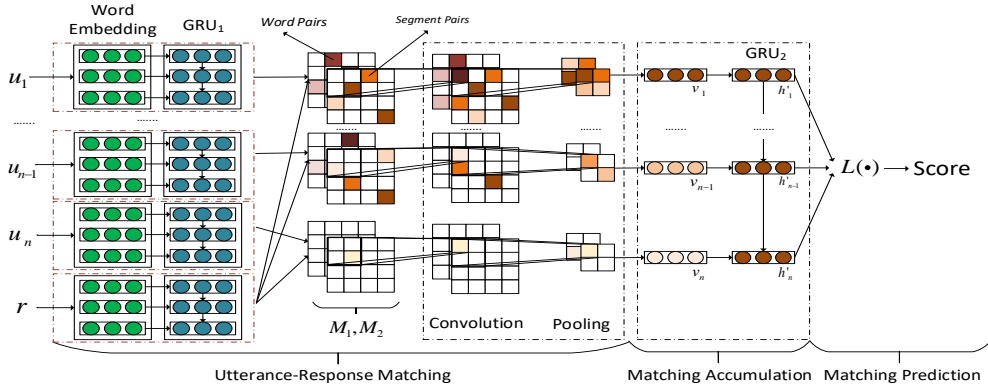

Figure 1: Architecture of SMN

et al., 2016; Serban et al., 2016a). Our work belongs to retrieval based methods, and we study context based response selection.

Early studies of retrieval based chatbots focus on response selection for single-turn conversation (Wang et al., 2013; Ji et al., 2014; Wang et al., 2015; Wu et al., 2016b). Recently, researchers begin to pay attention to multi-turn conversation. For example, Lowe et al. (2015) match a response with the literal concatenation of context utterances. Yan et al. (2016) concatenate context utterances with the input message as reformulated queries and perform matching with a deep neural network architecture. Zhou et al. (2016) improve multi-turn response selection with a multi-view model including an utterance view and a word view. Our model is different in that it matches a response with each utterance at first and accumulates matching information instead of sentences by a GRU, thus useful information for matching can be sufficiently retained.

## 3 Sequential Matching Network

### 3.1 Problem Formalization

Suppose that we have a data set $\mathcal{D} = \{(y_i, s_i, r_i)\}_{i=1}^N$, where $s_i = \{u_{i,1}, \ldots, u_{i,n_i}\}$ represents a conversation context with $\{u_{i,k}\}_{k=1}^{n_i}$ as utterances. $r_i$ is a response candidate and $y_i \in \{0, 1\}$ denotes a label. $y_i = 1$ means $r_i$ is a proper response for $s_i$, otherwise $y_i = 0$. Our goal is to learn a matching model $g(\cdot, \cdot)$ with $\mathcal{D}$. For any context-response pair $(s, r)$, $g(s, r)$ measures the matching degree between $s$ and $r$.

### 3.2 Model Overview

We propose a sequential matching network (SMN) to model $g(\cdot, \cdot)$. Figure 1 gives the architecture.

SMN first decomposes context-response matching into several utterance-response pair matching and then all pair matching is accumulated as a context based matching through a recurrent neural network. SMN consists of three layers. The first layer matches a response candidate with each utterance in the context on a word level and a segment level, and important matching information from the two levels is distilled by convolution and pooling and encoded in a matching vector. The matching vectors are then fed into the second layer where they are accumulated in the hidden states of a recurrent neural network with GRU following the chronological order of the utterances in the context. The third layer calculates the final matching score with the hidden states of the second layer.

SMN enjoys several advantages over the existing models. First, a response candidate can meet each utterance in the context at the very beginning, thus matching information in every utterance-response pair can be sufficiently extracted and carried to the final matching score with minimal loss. Second, information extraction from each utterance is conducted on different levels of granularity and under sufficient supervision from the response, thus semantic structures that are useful to response selection in each utterance can be well identified and extracted. Third, matching and utterance relationships are coupled rather than separately modeled, thus utterance relationships (e.g., order), as a kind of knowledge, can supervise the formation of the matching score.

By taking utterance relationships into account, SMN extends the "2D" matching that has proven effective in text pair matching for single-turn response selection to sequential "2D" matching for context based matching in response selection for multi-turn conversation. In the following sections,

we will describe details of the three layers.

### 3.3 Utterance-Response Matching

Given an utterance $u$ in a context $s$ and a response candidate $r$, the model looks up an embedding table and represents $u$ and $r$ as $\mathbf{U} = [e_{u,1}, \ldots, e_{u,n_u}]$ and $\mathbf{R} = [e_{r,1}, \ldots, e_{r,n_r}]$ respectively, where $e_{u,i}, e_{r,i} \in \mathbb{R}^d$ are the embeddings of the $i$-th word of $u$ and $r$ respectively. $\mathbf{U} \in \mathbb{R}^{d \times n_u}$ and $\mathbf{R} \in \mathbb{R}^{d \times n_r}$ are then used to construct a word-word similarity matrix $\mathbf{M}_1 \in \mathbb{R}^{n_u \times n_r}$ and a sequence-sequence similarity matrix $\mathbf{M}_2 \in \mathbb{R}^{n_u \times n_r}$ which are two input channels of a convolutional neural network (CNN). The CNN distills important matching information from the matrices and encodes the information into a matching vector $v$.

Specifically, $\forall i, j$, the $(i, j)$-th element of $\mathbf{M}_1$ is defined by

$$e_{1,i,j} = e_{u,i}^\top \cdot e_{r,j}. \tag{1}$$

$\mathbf{M}_1$ models the matching between $u$ and $r$ on a word level.

To construct $\mathbf{M}_2$, we first employ a GRU to transform $\mathbf{U}$ and $\mathbf{R}$ to hidden vectors. Suppose that $\mathbf{H}_u = [h_{u,1}, \ldots, h_{u,n_u}]$ are the hidden vectors of $\mathbf{U}$, then $\forall i, h_{u,i} \in \mathbb{R}^m$ is defined by

$$
\begin{aligned}
z_i &= \sigma(\mathbf{W_z} e_{u,i} + \mathbf{U_z} h_{u,i-1}) \\
r_i &= \sigma(\mathbf{W_r} e_{u,i} + \mathbf{U_r} h_{u,i-1}) \\
\widetilde{h}_{u,i} &= tanh(\mathbf{W_h} e_{u,i} + \mathbf{U_h}(r_i \odot h_{u,i-1})) \\
h_{u,i} &= z_i \odot \widetilde{h}_{u,i} + (1 - z_i) \odot h_{u,i-1},
\end{aligned} \tag{2}
$$

where $h_{u,0} = 0$, $z_i$ and $r_i$ are an update gate and a reset gate respectively, $\sigma(\cdot)$ is a sigmoid function, and $\mathbf{W_z}$, $\mathbf{W_h}$, $\mathbf{W_r}$, $\mathbf{U_z}$, $\mathbf{U_r}$, $\mathbf{U_h}$ are parameters. Similarly, we have $\mathbf{H}_r = [h_{r,1}, \ldots, h_{r,n_r}]$ as the hidden vectors of $\mathbf{R}$. Then, $\forall i, j$, the $(i, j)$-th element of $\mathbf{M}_2$ is defined by

$$e_{2,i,j} = h_{u,i}^\top \mathbf{A} h_{r,j}, \tag{3}$$

where $\mathbf{A} \in \mathbb{R}^{m \times m}$ is a linear transformation. $\forall i$, GRU models the sequential relationship and the dependency among words up to position $i$ and encodes the text segment until the $i$-th word to a hidden vector. Therefore, $\mathbf{M}_2$ models the matching between $u$ and $r$ on a segment level.

$\mathbf{M}_1$ and $\mathbf{M}_2$ are then processed by a CNN to form $v$. $\forall f = 1, 2$, CNN regards $\mathbf{M}_f$ as an input channel, and alternates convolution and max-pooling operations. Suppose that $z^{(l,f)} =$

$\left[ z_{i,j}^{(l,f)} \right]_{I^{(l,f)} \times J^{(l,f)}}$ denotes the output of feature maps of type-$f$ on layer-$l$, where $z^{(0,f)} = \mathbf{M}_f$, $\forall f = 1, 2$. On the convolution layer, we employ a 2D convolution operation with a window size $r_w^{(l,f)} \times r_h^{(l,f)}$, and define $z_{i,j}^{(l,f)}$ as

$$z_{i,j}^{(l,f)} = \sigma\left(\sum_{f'=0}^{F_{l-1}} \sum_{s=0}^{r_w^{(l,f)}} \sum_{t=0}^{r_h^{(l,f)}} \mathbf{W}_{s,t}^{(l,f)} \cdot z_{i+s,j+t}^{(l-1,f')} + \mathbf{b}^{l,k}\right), \tag{4}$$

where $\sigma(\cdot)$ is a ReLU, $\mathbf{W}^{(l,f)} \in \mathbb{R}^{r_w^{(l,f)} \times r_h^{(l,f)}}$ and $\mathbf{b}^{l,k}$ are parameters, and $F_{l-1}$ is the number of feature maps on the $(l-1)$-th layer. A max pooling operation follows a convolution operation and can be formulated as

$$z_{i,j}^{(l,f)} = \max_{p_w^{(l,f)} > s \geq 0} \max_{p_h^{(l,f)} > t \geq 0} z_{i+s,j+t}, \tag{5}$$

where $p_w^{(l,f)}$ and $p_h^{(l,f)}$ are the width and the height of the 2D pooling respectively. The output of the final feature maps are concatenated and mapped to a low dimensional space with a linear transformation as the matching vector $v \in \mathbb{R}^q$.

From Equation (1), (3), (4), and (5), we can see that by learning word embedding and parameters of GRU from training data, words or segments in an utterance that are useful to recognize the appropriateness of a response may have high similarity with some words or segments in the response and result in high value areas in the similarity matrices. These areas will be transformed and selected by convolution and pooling operations and carry the important information in the utterance to the matching vector. This is how our model identifies important information in context and leverage it in matching under the supervision of the response. We consider multiple channels because we want to capture important matching information on multiple levels of granularity of text.

### 3.4 Matching Accumulation

Suppose that $[v_1, \ldots, v_n]$ is the output of the first layer (corresponding to $n$ pairs), at the second layer, a GRU takes $[v_1, \ldots, v_n]$ as an input and encodes the matching sequence into its hidden states $H_m = [h'_1, \ldots, h'_n] \in \mathbb{R}^{q \times n}$ with a detailed parameterization similar to Equation (2). This layer has two functions: (1) it models the dependency and the temporal relationship of utterances in the context; (2) it leverages the temporal relationship to supervise the accumulation of the pair matching as a context based matching. Moreover, from

Equation (2), we can see that the reset gate (i.e., $r_i$) and the update gate (i.e., $z_i$) control how much information from the previous hidden state and the current input flows to the current hidden state, thus important matching vectors (corresponding to important utterances) can be accumulated while noise in the vectors can be filtered out.

### 3.5 Matching Prediction and Learning

With $[h'_1, \ldots, h'_n]$, we define $g(s, r)$ as

$$g(s, r) = softmax(\mathbf{W_2} L[h'_1, \ldots, h'_n] + \mathbf{b_2}), \quad (6)$$

where $\mathbf{W_2}$ and $\mathbf{b_2}$ are parameters. We consider three parameterizations for $L[h'_1, \ldots, h'_n]$: (1) only the last hidden state is used. Then $L[h'_1, \ldots, h'_n] = h'_n$. (2) the hidden states are linearly combined. Then, $L[h'_1, \ldots, h'_n] = \sum_{i=1}^{n} w_i h'_i$, where $w_i \in \mathbb{R}$. (3) we follow (Yang et al., 2016) and employ an attention mechanism to combine the hidden states. Then, $L[h'_1, \ldots, h'_n]$ is defined as

$$t_i = tanh(\mathbf{W_{1,1}} h_{u_i, n_u} + \mathbf{W_{1,2}} h'_i + \mathbf{b_1}),$$
$$\alpha_i = \frac{exp(t_i^\top t_s)}{\sum_i (exp(t_i^\top t_s))},$$
$$L[h'_1, \ldots, h'_n] = \sum_{i=1}^{n} \alpha_i h'_i, \quad (7)$$

where $\mathbf{W_{1,1}} \in \mathbb{R}^{q \times m}, \mathbf{W_{1,2}} \in \mathbb{R}^{q \times q}$ and $\mathbf{b_1} \in \mathbb{R}^q$ are parameters. $h'_i$ and $h_{u_i, n_u}$ are the $i$-th matching vector and the final hidden state of the $i$-th utterance respectively. $t_s \in \mathbb{R}^q$ is a virtual context vector which is randomly initialized and jointly learned in training.

Both (2) and (3) aim to learn weights for $\{h'_1, \ldots, h'_n\}$ from training data and highlight the effect of important matching vectors in the final matching. The difference is that weights in (2) are static, because the weights are totally determined by the positions of utterances, while weights in (3) are dynamically computed by the matching vectors and utterance vectors. We denote our model with the three parameterizations of $L[h'_1, \ldots, h'_n]$ as $SMN_{last}$, $SMN_{static}$, and $SMN_{dynamic}$, and empirically compare them in experiments.

We learn $g(\cdot, \cdot)$ by minimizing cross entropy with $\mathcal{D}$. Let $\Theta$ denote the parameters of SMN, then the objective function $\mathcal{L}(\mathcal{D}, \Theta)$ of learning can be formulated as

$$-\sum_{i=1}^{N} [y_i log(g(s_i, r_i)) + (1 - y_i) log(1 - g(s_i, r_i))]. \quad (8)$$

## 4 Response Candidate Retrieval

In practice of a retrieval based chatbot, to apply the matching approach to response selection, one needs to retrieve a bunch of response candidates from an index beforehand. While candidate retrieval is not the focus of the paper, it is an important step in a real system. In this work, we exploit a heuristic method to obtain response candidates from the index. Given a message $u_n$ with $\{u_1, \ldots, u_{n-1}\}$ utterances in its previous turns, we extract top 5 keywords from $\{u_1, \ldots, u_{n-1}\}$ based on their tf-idf scores[1] and expand $u_n$ with the keywords. Then we send the expanded message to the index and retrieve response candidates using the inline retrieval algorithm of the index. Finally, we use $g(s, r)$ to re-rank the candidates and return the top one as a response to the context.

## 5 Experiments

We tested our model on a public English data set and a Chinese data set we publish with this paper.

### 5.1 Ubuntu Corpus

The English data set is the Ubuntu Corpus (Lowe et al., 2015) which contains multi-turn dialogues collected from chat logs of Ubuntu Forum. The data set consists of 1 million context-response pairs for training, 0.5 million pairs for validation, and 0.5 million pairs for test. Positive responses are true responses from human, and negative ones are randomly sampled. The ratio of the positive and the negative is 1:1 in training, and 1:9 in validation and test. We used the copy shared by Xu et al. (2016) [2] in which numbers, urls, and paths are replaced by special placeholders. We followed (Lowe et al., 2015) and employed recall at position $k$ in $n$ candidates ($R_n@k$) as evaluation metrics.

### 5.2 Douban Conversation Corpus

Ubuntu Corpus is a domain specific data set, and response candidates are obtained from negative sampling without human judgment. To further verify the efficacy of our model, we created a new data set with open domain conversations, namely Douban Conversation Corpus. Response candidates in the test set of Douban Conversation Corpus are collected following the procedure of a re-

---

[1]Tf is word frequency in the context, while idf is calculated using the entire index.

[2]https://www.dropbox.com/s/2fdn26rj6h9bpvl/ubuntudata.zip?dl=0

trieval based chatbot and are labeled by human judges. Douban Conversation Corpus simulates the real scenario of a retrieval based chatbot, and we publish it to research communities to facilitate the research of multi-turn response selection.

Specifically, we crawled 1.1 million dyadic dialogues (conversation between two persons) longer than 2 turns from Douban group[3] which is a popular social networking service in China. From the data, we randomly sampled 0.5 million dialogues for creating a training set, 25 thousand dialouges for creating a validation set, and $1,000$ dialogues for creating a test set, and made sure that there is no overlap among the three sets. For each dialogue in training and validation, we took the last turn as a positive response for the previous turns as a context and randomly sampled another response from the 1.1 million data as a negative response. There are 1 million context-response pairs in the training set and 50 thousand pairs in the validation set.

To create the test set, we first crawled 15 million post-reply pairs from Sina Weibo[4] which is the largest microblogging service in China and indexed the pairs with Lucene[5]. We took the last turn of each Douban dyadic dialogue in the test set as a message, retrieved 10 response candidates from the index following the method in Section 4, and finally formed a test set with $10,000$ context-response pairs. We recruited three labelers to judge if a candidate is a proper response to the context. A proper response means the response can naturally reply to the message given the whole context. Each pair received three labels and the majority of the labels were taken as the final decision. Table 2 gives the statistics of the three sets. Note that the Fleiss' kappa (Fleiss, 1971) of the labeling is $0.41$, which indicates that the three labelers reached a relatively high agreement.

Besides $R_n@k$s, we also followed the convention of information retrieval and employed mean average precision (MAP) (Baeza-Yates et al., 1999), mean reciprocal rank (MRR) (Voorhees et al., 1999), and precision at position 1 (P@1) as evaluation metrics. We did not calculate $R_2@1$ because in Douban corpus one context could have more than one correct responses, and we have to randomly sample one for $R_2@1$, which may bring bias to evaluation. When using the labeled set,

| | train | val | test |
|---|---|---|---|
| # context-response pairs | 1M | 50k | 10k |
| # candidates per context | 2 | 2 | 10 |
| # positive candidates per context | 1 | 1 | 1.18 |
| Min. # turns per context | 3 | 3 | 3 |
| Max. # turns per context | 98 | 91 | 45 |
| Avg. # turns per context | 6.69 | 6.75 | 6.45 |
| Avg. # words per utterance | 18.56 | 18.50 | 20.74 |

Table 2: Statistics of Douban Conversation Corpus

we removed conversations with all negative responses or all positive responses, as models make no difference on them. There are $6,670$ context-response pairs left in the test set.

## 5.3 Baseline

We considered the following baselines:

**Basic models**: models in (Lowe et al., 2015) and (Kadlec et al., 2015) including TF-IDF, RNN, CNN, LSTM and BiLSTM.

**Multi-view**: the model proposed by Zhou et al. (2016) that utilizes a hierarchical recurrent neural network to model utterance relationships.

**Deep learning to respond (DL2R)**: the model proposed by Yan et al. (2016) that reformulates the message with other utterances in the context.

**Advanced single-turn matching models**: since BiLSTM does not represent the state-of-the-art matching model, we concatenated the utterances in a context and matched the long text with a response candidate using more powerful models including MV-LSTM (Wan et al., 2016) (2D matching), Match-LSTM (Wang and Jiang, 2015), Attentive-LSTM (Tan et al., 2015) (two attention based models), and Multi-Channel which is described in Section 3.3. Multi-Channel is a simple version of our model without considering utterance relationships. We also appended the top 5 tf-idf words in context to the input message, and computed the score between the expanded message and a response with Multi-Channel, denoted as Multi-Channel$_{exp}$.

## 5.4 Parameter Tuning

For baseline models, if their results are available in the existing literatures (e.g., those on the Ubuntu corpus), we just copied the numbers, otherwise we implemented the models following the settings in the literatures. All models were implemented using Theano (Theano Development Team, 2016). Word embeddings were initialized by the results of word2vec (Mikolov et al., 2013) which ran on the training data, and the dimensionality of word vectors is 200. For Multi-Channel and layer one of

---

[3] https://www.douban.com/group
[4] http://weibo.com/
[5] https://lucenenet.apache.org/

| | Ubuntu Corpus | | | | Douban Conversation Corpus | | | | | |
|---|---|---|---|---|---|---|---|---|---|---|
| | $R_2@1$ | $R_{10}@1$ | $R_{10}@2$ | $R_{10}@5$ | MAP | MRR | P@1 | $R_{10}@1$ | $R_{10}@2$ | $R_{10}@5$ |
| TF-IDF | 0.659 | 0.410 | 0.545 | 0.708 | 0.331 | 0.359 | 0.179 | 0.095 | 0.172 | 0.405 |
| RNN | 0.768 | 0.403 | 0.547 | 0.819 | 0.390 | 0.422 | 0.208 | 0.011 | 0.223 | 0.589 |
| CNN | 0.848 | 0.549 | 0.684 | 0.896 | 0.417 | 0.440 | 0.226 | 0.012 | 0.252 | 0.647 |
| LSTM | 0.901 | 0.638 | 0.784 | 0.949 | 0.485 | 0.527 | 0.320 | 0.187 | 0.343 | 0.720 |
| BiLSTM | 0.895 | 0.630 | 0.780 | 0.944 | 0.479 | 0.514 | 0.313 | 0.184 | 0.330 | 0.716 |
| Multi-View | 0.908 | 0.662 | 0.801 | 0.951 | 0.505 | 0.543 | 0.342 | 0.202 | 0.350 | 0.729 |
| DL2R | 0.899 | 0.626 | 0.783 | 0.944 | 0.488 | 0.527 | 0.330 | 0.193 | 0.342 | 0.705 |
| MV-LSTM | 0.906 | 0.653 | 0.804 | 0.946 | 0.498 | 0.538 | 0.348 | 0.202 | 0.351 | 0.710 |
| Match-LSTM | 0.904 | 0.653 | 0.799 | 0.944 | 0.500 | 0.537 | 0.345 | 0.202 | 0.348 | 0.720 |
| Attentive-LSTM | 0.903 | 0.633 | 0.789 | 0.943 | 0.495 | 0.523 | 0.331 | 0.192 | 0.328 | 0.718 |
| Multi-Channel | 0.904 | 0.656 | 0.809 | 0.942 | 0.506 | 0.543 | 0.349 | 0.203 | 0.351 | 0.709 |
| Multi-Channel$_{exp}$ | 0.714 | 0.368 | 0.497 | 0.745 | 0.476 | 0.515 | 0.317 | 0.179 | 0.335 | 0.691 |
| SMN$_{last}$ | **0.923** | **0.723** | **0.842** | **0.956** | **0.526** | **0.571** | **0.392** | **0.236** | **0.387** | 0.729 |
| SMN$_{static}$ | **0.927** | **0.725** | **0.838** | **0.962** | **0.523** | **0.572** | **0.387** | **0.228** | **0.387** | 0.734 |
| SMN$_{dynamic}$ | **0.926** | **0.726** | **0.847** | **0.961** | **0.529** | **0.569** | **0.395** | **0.233** | **0.396** | 0.724 |

Table 3: Evaluation results on the two data sets. Numbers in bold mean that the improvement is statistically significant compared with the best baseline.

our model, we set the dimensionality of the hidden states of GRU as 200. We tuned the window size of convolution and pooling in $\{(2,2),(3,3)(4,4)\}$ and chose $(3,3)$ finally. The number of feature maps is 8. In layer two, we set the dimensionality of matching vectors and the hidden states of GRU as 50. The parameters were updated by stochastic gradient descent with Adam algorithm (Kingma and Ba, 2014) on a single Tesla K80 GPU. The initial learning rate is 0.001, and the parameters of Adam, $\beta_1$ and $\beta_2$ are 0.9 and 0.999 respectively. We employed early-stopping as a regularization strategy. Models were trained in mini-batches with a batch size 200, and maximum utterance length is 50. We set the maximum context length (i.e., number of utterances) as 10, because performance of models does not get improved on contexts longer than 10 (details are shown in the supplementary material). We padded zeros if the number of utterances in a context is less than 10, otherwise we kept the last 10 utterances.

## 5.5 Evaluation Results

Table 3 shows the evaluation results on the two data sets. Our models outperform baselines greatly in terms of all metrics on both data sets, and the improvements are statistically significant (t-test with $p$-value $\leq 0.01$, except $R_{10}@5$ on Douban Corpus). Even the state-of-the-art single-turn matching models perform much worse than our models. The results demonstrate that one cannot neglects utterance relationships and simply perform multi-turn response selection by concatenating utterances together. Our models achieve significant improvements over Multi-View, which justified our "matching first" strategy. DL2R is

worse than our models, indicating that utterance reformulation with heuristic rules is not a good method to utilize context information. $R_n@k$s are low on Douban corpus as there are multiple correct candidates for a context (e.g., if there are 3 correct responses, then the maximum $R_{10}@1$ is 0.33). SMN$_{dynamic}$ is only slightly better than SMN$_{static}$ and SMN$_{last}$. The reason might be that GRU can select useful signals from the matching sequence and accumulate them in the final state with its gate mechanism, thus the efficacy of attention mechanism is not obvious for the task.

## 5.6 Further Analysis

**Visualization**: we visualize the similarity matrices and the gates of GRU in layer two using an example from the Ubuntu corpus to further clarify how our model identifies important information in the context and how it selects important matching vectors with the gate mechanism of GRU as described in Section 3.3 and Section 3.4. The example is $\{u_1$: *how can unzip many rar ( _number_ for example ) files at once; $u_2$: sure you can do that in bash; $u_3$: okay how? $u_4$: are the files all in the same directory? $u_5$: yes they all are; $r$: then the command glebihan should extract them all from/to that directory*$\}$. It is from the test set and our model successfully ranked the correct response to the top position. Due to space limitation, we only visualized $M_1$, $M_2$ and the update gate (i.e. $z$) in Figure 2. Other pieces of our model are shown in the supplementary material. We can see that in $u_1$ important words including "unzip", "rar", "files" are recognized and carried to matching by "command", "extract", and "directory" in $r$, while $u_3$ is almost useless and thus little infor-

| | Ubuntu Corpus | | | | Douban Conversation Corpus | | | | | |
|---|---|---|---|---|---|---|---|---|---|---|
| | $R_2@1$ | $R_{10}@1$ | $R_{10}@2$ | $R_{10}@5$ | MAP | MRR | P@1 | $R_{10}@1$ | $R_{10}@2$ | $R_{10}@5$ |
| $\text{Replace}_M$ | 0.905 | 0.661 | 0.799 | 0.950 | 0.503 | 0.541 | 0.343 | 0.201 | 0.364 | 0.729 |
| $\text{Replace}_A$ | 0.918 | 0.716 | 0.832 | 0.954 | 0.522 | 0.565 | 0.376 | 0.220 | 0.385 | 0.727 |
| Only $M_1$ | 0.919 | 0.704 | 0.832 | 0.955 | 0.518 | 0.562 | 0.370 | 0.228 | 0.371 | 0.737 |
| Only $M_2$ | 0.921 | 0.715 | 0.836 | 0.956 | 0.521 | 0.565 | 0.382 | 0.232 | 0.380 | 0.734 |
| $\text{SMN}_{last}$ | 0.923 | 0.723 | 0.842 | 0.956 | 0.526 | 0.571 | 0.392 | 0.236 | 0.387 | 0.729 |

Table 4: Evaluation results of model ablation.

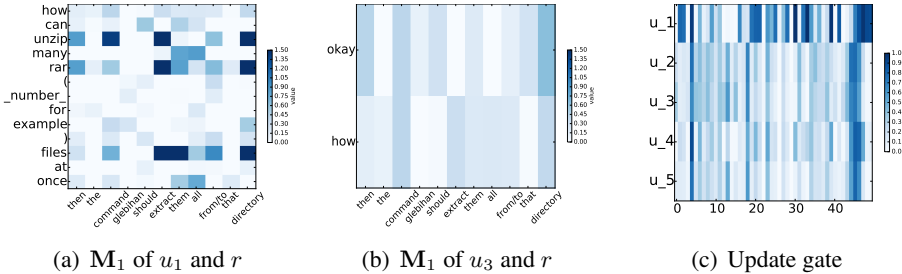

(a) $\mathbf{M}_1$ of $u_1$ and $r$ (b) $\mathbf{M}_1$ of $u_3$ and $r$ (c) Update gate

Figure 2: Model visualization. Darker areas mean larger value.

mation is extracted from it. $u_1$ is crucial to response selection and nearly all information from $u_1$ and $r$ flows to the hidden state of GRU, while other utterances are less informative and the corresponding gates are almost "closed" to keep the information from $u_1$ and $r$ until the final state.

**Model ablation**: we investigate the effect of different parts of SMN by removing them one by one from $\text{SMN}_{last}$, shown in Table 4. First, replacing the multi-channel "2D" matching with a neural tensor network (NTN) (Socher et al., 2013) (denoted as $\text{Replace}_M$) makes the performance drop dramatically. This is because NTN only matches a pair by an utterance vector and a response vector and loses important information in the pair. Together with the visualization, we can conclude that "2D" matching plays a key role in the "matching first" strategy as it captures the important matching information in each pair with minimal loss. Second, the performance slightly drops when replacing the GRU for matching accumulation with a multi-layer perceptron (denoted as $\text{Replace}_A$). This indicates that utterance relationships are useful. Finally, we left only one channel in matching and found that $\mathbf{M}_2$ is a little more powerful than $\mathbf{M}_1$ and we achieve the best results with both of them (except on $R_{10}@5$ on Douban Corpus).

**Context length**: we study how our model ($\text{SMN}_{last}$) performs across the length of contexts. Figure 3 shows the comparison on MAP in different length intervals on the Douban corpus. Our model consistently performs better than the baselines, and when contexts become longer, the gap becomes larger. The results demonstrate that our model can well capture the dependencies, espe-

cially long dependencies, among utterances in contexts. We give the comparisons on other metrics in our supplementary material.

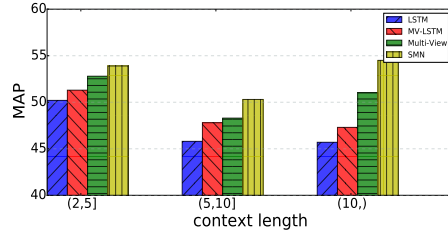

Figure 3: Comparison across context length

**Retrieval v.s. Generation**: we compared SMN with a state-of-the-art response generation model VHERD (Serban et al., 2016b) which was trained using $\mathcal{D}$ on the Douban corpus. We conducted a side-by-side human comparison on the top one responses of the two models for each context in the test set. The result is that SMN wins on 238 examples, loses on 207 examples, and is comparable with VHRED on the remaining 555 examples. This indicates that a retrieval based chatbot with SMN can provide a better experience than the state-of-the-art generation model in practice.

# 6 Conclusion and Future Work

We present a new context based model for multi-turn response selection in retrieval-based chatbots. Experiment results on public data sets show that the model can significantly outperform the state-of-the-art methods. Besides, we publish the first human labeled multi-turn response selection data set to research communities. In the future, we are going to study how to model logical consistency of responses and improve candidate retrieval (see supplementary material).

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
