# Peer review of "Sequential Matching Network: A New Architecture for Multi-turn Response Selection in Retrieval-Based Chatbots"

_ACL 2017 — decision unknown_

[Official Review · Reviewer 1 · rating 4 · confidence 2]
soundness 5 · originality 3 · clarity 4 · impact 3 · substance 4 · appropriateness 5 · meaningful comparison 5 · presentation format Poster

- Strengths:

Relatively clear description of context and structure of proposed approach.
Relatively complete description of the math. Comparison to an extensive set of
alternative systems.

- Weaknesses:

Weak results/summary of "side-by-side human" comparison in Section 5. Some
disfluency/agrammaticality.

- General Discussion:

The article proposes a principled means of modeling utterance context,
consisting of a sequence of previous utterances. Some minor issues:

1. Past turns in Table 1 could be numbered, making the text associated with
this table (lines 095-103) less difficult to ingest. Currently, readers need to
count turns from the top when identifying references in the authors'
description, and may wonder whether "second", "third", and "last" imply a
side-specific or global enumeration.

2. Some reader confusion may be eliminated by explicitly defining what
"segment" means in "segment level", as occurring on line 269. Previously, on
line 129, this seemingly same thing was referred to as "a sequence-sequence
[similarity matrix]". The two terms appear to be used interchangeably, but it
is not clear what they actually mean, despite the text in section 3.3. It seems
the authors may mean "word subsequence" and "word subsequence to word
subsequence", where "sub-" implies "not the whole utterance", but not sure.

3. Currently, the variable symbol "n" appears to be used to enumerate words in
an utterance (line 306), as well as utterances in a dialogue (line 389). The
authors may choose two different letters for these two different purposes, to
avoid confusing readers going through their equations.

4. The statement "This indicates that a retrieval based chatbot with SMN can
provide a better experience than the state-of-the-art generation model in
practice." at the end of section 5 appears to be unsupported. The two
approaches referred to are deemed comparable in 555 out of 1000 cases, with the
baseline better than the proposed method in 238 our of the remaining 445 cases.
The authors are encouraged to assess and present the statistical significance
of this comparison. If it is weak, their comparison permits to at best claim
that their proposed method is no worse (rather than "better") than the VHRED
baseline.

5. The authors may choose to insert into Figure 1 the explicit "first layer",
"second layer" and "third layer" labels they use in the accompanying text.

6.  Their is a pervasive use of "to meet" as in "a response candidate can meet
each utterace" on line 280 which is difficult to understand.

7. Spelling: "gated recurrent unites"; "respectively" on line 133 should be
removed; punctuation on line 186 and 188 is exchanged; "baseline model over" ->
"baseline model by"; "one cannot neglects".